# In Situ Preparation of Tannic Acid-Modified Poly(*N*-isopropylacrylamide) Hydrogel Coatings for Boosting Cell Response

**DOI:** 10.3390/pharmaceutics16040538

**Published:** 2024-04-13

**Authors:** Jufei Xu, Xiangzhe Liu, Pengpeng Liang, Hailong Yuan, Tianyou Yang

**Affiliations:** 1Department of Pharmacy, Air Force Medical Center, PLA, Air Force Medical University, Beijing 100142, China; jfxu@alu.ruc.edu.cn; 2School of Life Science and Technology, Henan Institute of Science and Technology, Xinxiang 453003, China; lxz20110@163.com; 3University of Chinese Academy of Sciences, Beijing 100049, China; 4School of Materials Science and Engineering, Zhengzhou University, Zhengzhou 450001, China; liangpp2024@126.com

**Keywords:** modified poly(*N*-isopropylacrylamide) coatings, coating properties, protein behaviors, cell response

## Abstract

The improvement of the capability of poly(*N*-isopropylacrylamide) (PNIPAAm) hydrogel coating in cell adhesion and detachment is critical to efficiently prepare cell sheets applied in cellular therapies and tissue engineering. To enhance cell response on the surface, the amine group-modified PNIPAAm (PNIPAAm-APTES) nanohydrogels were synthesized and deposited spontaneously on tannic acid (TA)-modified polyethylene (PE) plates. Subsequently, TA was introduced onto PNIPAAm-APTES nanohydrogels to fabricate coatings composed of TA-modified PNIPAAm-APTES (PNIPAAm-APTES-TA). Characterization techniques, including TEM, SEM, XPS, and UV-Vis spectroscopy, confirmed the effective deposition of hydrogels of PNIPAAm as well as the morphologies, content of chemical bonding-TA, and stability of various coatings. Importantly, the porous hydrogel coatings exhibited superhydrophilicity at 20 °C and thermo-responsive behavior. The fluorescence measurement demonstrated that the coating’s stability effectively regulated protein behavior, influencing cell response. Notably, cell response tests revealed that even without precise control over the chain length/thickness of PNIPAAm during synthesis, the coatings enhanced cell adhesion and detachment, facilitating efficient cell culture. This work represented a novel and facile approach to preparing bioactive PNIPAAm for cell culture.

## 1. Introduction

Over the past decades, materials and their interface designs for excellent cell response have attracted extensive attention for their potential application in many fields with respect to tissue engineering and cellular therapies [1,2,3,4]. The interface of materials, which is responsible for the initial interactions between a cell and a foreign substrate, plays a crucial role in determining cell response to materials, including aspects such as compatibility, cell adhesion, proliferation, and subsequent cell fate [5,6]. Additionally, if the interfacial environment is capable of promoting spontaneous cell detachment during the detachment phase, to a large extent, the damage to the cell by the detachment force is diminished. Consequently, preserving the multifunctionality of cells, which includes their ability to differentiate, migrate, and communicate with other cells, is ensured [7,8,9]. The development of advanced materials and rational fabrication of interfaces is vital to satisfying the demands of the related diverse fields.

Hydrogel modification interfaces represent an interesting format for cell culture due to their capacity to imitate the extracellular matrix (ECM) [10]. The native ECM that surrounds the cell directs cellular functions and modulates cell behaviors [11]. Hydrogels not only provide an aqueous microenvironment similar to the natural ECM but also mimic key characteristics of the ECM, such as providing suitable sites for cell adhesion and facilitating the diffusion of growth factors [12,13,14,15]. Poly(*N*-isopropylacrylamide) (PNIPAAm), a well-known thermo-sensitive material, exhibits excellent biocompatibility and thermo-responsiveness near physiological temperatures (~37 °C). Beyond these uses in drug delivery through hydrogels, PNIPAAm and its copolymers have been utilized for in vitro cell culture, with surfaces modified by various polymerization techniques [16]. The thermo-sensitive properties of PNIPAAm have been leveraged to regulate cell adhesion and detachment, thereby enhancing cell viability and culture efficiency. Okano et al. reported that approximately 100% of cells detached from a PNIPAAm-modified surface within 60 min by lowering the temperature [17]. The thickness of the PNIPAAm coating significantly affects cell adhesion as the hydration of molecular chains of varying lengths at the outermost layer differs [18]. Thus, strict control of the quality of the PNIPAAm coating is essential to improve cell adhesion and detachment. However, the achievement of this goal is complex due to polymerization techniques such as copolymerization and controlled polymerization [16,19]. In addition, to improve cell detachment efficiency, additional processes, such as the preparation of porous fiber film, are usually required, and the adhesion of the surface to the cell is more or less sacrificed. Therefore, the fabrication of the PNIPAAm coating appeals to novel and facile approaches to enhance cell adhesion without sacrificing detachment ability.

Tannic acid (TA) is a hydrolyzable high-molecular polyphenol compound originating from plants [20]. It has exhibited some abilities in the regulation of the behaviors of various bioactive factors involved in cell culture. Generally, these bioactive factors are some components of ECM or are secreted by cells. TA can interact with various materials via hydrogen bonding, π-π stacking, electrostatic interactions, coordination bonding, and hydrophobic interactions because of the phenol hydroxyl and phenyl groups in its chemical structure [21,22,23]. Thereby, through diverse interactions with bioactive factors, TA and its complexes have been utilized to control some bioactive factors to modulate cell response [24]. For example, Revzin et al. reported that ultrathin TA-coated heparin microgels were able to capture about two times the basic fibroblast growth factor (BFGF) compared to bare heparin microgels. The BFGF release was prolonged by 1–2 d, inducing human pluripotent stem cells to express a higher level of pluripotency [25]. Therefore, TA was incorporated into PNIPAAm coatings, potentially endowing the coating with the ability to mediate protein behaviors and elevate cell adhesion without the need for rigorous tailoring of coating thickness. However, achieving a stable TA-modified PNIPAAm coating through the direct application of TA is challenging due to the high hydration of the outermost layer of PNIPAAm and the deprotonation of TA.

In this work, TA-modified PNIPAAm (PNIPAAm-APTES-TA) hydrogel coatings, characterized by porous morphology and stable structure, were fabricated via in situ deposition, as depicted in Figure 1. Amino-terminated PNIPAAm (PNIPAAm-APTES) variants with varying amine group content were synthesized and spontaneously deposited on TA/Fe^3+^-modified polyethylene (PE) plates. Subsequently, TA was introduced onto PNIPAAm-APTES to prepare PNIPAAm-APTES-TA coatings. Herein, PNIPAAm-APTES-TA coatings with varying levels of chemically bonded TA were prepared by modulating the amino content. The stability of various PNIPAAm coatings was assessed, and the optimum ones were selected. The effects of the coatings on protein behavior and cell response were also investigated.

## 2. Materials and Methods

### 2.1. Materials

Tissue-culture polyethylene (CPE) and ordinary polyethylene (PE) were provided by NEST Biotechnology Co., Ltd., Wuxi, China. Hydroxyl-terminated poly(*N*-isopropylacrylamide) (PNIPAAm-OH) was prepared in our lab (Appendix A). Tannic acid (TA, ≥98%), rhodamine B isothiocyanate (BRITC, 99%), 3-aminopropylthiethoxysilane (APTES, 98%), alkaline phosphatase (ALP, 98%), an ALP assay kit, and DMEM were purchased from Sinopharm Chemical Reagent Co., Ltd., Shanghai, China. Mouse fibroblast cells (L929 cells), phosphate buffer saline (PBS, 99.9%, pH = 7.4), and fetal bovine serum (FBS, 97%) were obtained from Guangzhou Oricell Biotechnology Co., Ltd., Guangzhou, China. Calcein acetoxymethyl ester (Calcein AM) and propidium iodide (PI) were obtained from Solarbio, Beijing, China. Carbon dioxide (CO_2_, 99.999%) was supplied by Guangzhou Spectral Source Gas Co., Ltd., Guangzhou, China.

### 2.2. Amino-Terminated PNIPAAm-Modified Surface

Amino-terminated poly(*N*-isopropylacrylamide) (PNIPAAm-APTES) was synthesized by a condensation reaction, with the elimination of water between hydroxyl-terminated PNIPAAm (PNIPAAm-OH) and 3-aminopropylthiethoxysilane (APTES). Briefly, APTES was dissolved into a 90% (*v*:*v*) ethanol solution to prepare 3 wt% APTES solution, with mild stirring for 10 min at room temperature. The different amounts of PNIPAAm-OH were added rapidly into the APTES solution with vigorous magnetic stirring. After reaction for 30 min at room temperature, the pre-washed TA/Fe^3+^-modified PE plates (Appendix A depicts the preparation of the TA/Fe^3+^-modified PE plates) were put into the above mixture for another 4 h at 70 °C. The modified PE plates were completely washed with ethanol at 37 °C to remove residual reagents. The preparation condition of diverse PNIPAAm-APTES coatings (as a control) is listed in detail in Appendix A. Figure 1a,b illustrates the synthesis mechanism of PNIPAAm-APTES and in situ deposition of PNIPAAm-APTES on TA/Fe^3+^-modified PE plates, respectively.

### 2.3. Preparation of TA-Modified PNIPAAm-APTES Hydrogel Coatings

The preparation of the TA-modified PNIPAAm-APTES (PNIPAAm-APTES-TA) hydrogel coatings on PE plates was as follows: 12.5 mg of TA (pH value of TA was adjusted to 7.8 by Tris buffer) was sputter-coated on PNIPAAm-APTES hydrogel coatings that were used for modified PE plates. After reaction for another 4 h, PNIPAAm-APTES-TA-modified PE plates were obtained by lyophilization. In this process, part of the TA may react with the amine group of PNIPAAm-APTES via a Michael addition/Schiff base reaction to chemically graft TA on PNIPAAm-APTES (Figure 1a). The preparation condition of various PNIPAAm-APTES-TA coatings is detailed in Table 1. The process of TA-modified PNIPAAm-APTES is schematically illustrated in Figure 1b.

X-ray photoelectron spectroscopy (XPS, Thermo Fisher Scientific Inc., Waltham, MA, USA) was performed to confirm the chemical components and content of various coatings. The shape of the PNIPAAm-APTES particle and the interface morphologies of PNIPAAm-APTES-TA were characterized by transmission electron microscopy (TEM, TF20) and scanning electron microscopy (SEM) (S4800, Hitachi, Tokyo, Japan), respectively. The stability of the TA-modified hydrogel coatings was determined by monitoring the absorbance of the released TA in PBS at pH 7.4 using a UV-Vis spectrophotometer (U2550, Shimadzu, Kyoto, Japan). Static water contact angle (WCA) was measured using a contact angle meter with both a camera and temperature controller (SZ-CAMC33, SUNZERN, Shanghai, China) to characterize wettability.

### 2.4. Thermo-Sensitive Behaviors of Different Coatings

The volume phase transformation temperature is an important feature of the thermo-responsive polymer. To confirm their thermo-responsiveness, the hydrogel coatings were peeled from the PE surface and then dispersed into PBS to prepare a transparent solution. The solution was heated from 20 to 37 °C at the speed of 0.1 °C/min. The changes in turbidity for every solution were recorded by an optical camera to confirm the range of the volume phase transformation temperature. Dynamic light scanning (DLS) measurement was performed to monitor the hydrodynamic diameter, D_h_, of PNIPAAm-APTES-TA hydrogels and their variation with temperature, further characterizing the thermo-responsive capability of the different hydrogel coatings at themicro level. Owing to the dramatic change in volume near the volume phase transformation temperature, the thermo-sensitive hydrogel coating swelled or collapsed significantly, which caused an evident change in weight. Thus, the swelling rate and its change could be used to quantitatively reflect thermo-sensitive behavior. Briefly, the samples were immersed in PBS at the designated temperature for 24 h and were dried under vacuum. The dried samples were soaked in PBS at the same temperature as the ones mentioned above for 24 h. The weight of the completely swollen sample was measured. The swelling rate was calculated by the weight of the completely swollen sample divided by the weight of the dry sample. The lower critical solution temperature (LCST) of PNIPAAm-APTES-TA was measured using a UV-Vis spectrophotometer to determine the thermo-responsive behavior.

### 2.5. Protein Behaviors on Various Coatings

To confirm protein adsorption, the pre-labeled FBS with BRITC (BRITC-FBS) (0.15 g/mL) was added into the centrifuge tube that contained pristine PE and various surface-modified PE and then incubated at 37 °C for 0, 1, 3, 5, 12, and 24 h. After incubation, every specimen was centrifuged for 30 s at a speed of 100× *g*. All supernatants were collected for further analysis using a fluorophotometer (RF-6000, Shimadzu, Kyoto, Japan). According to the fluorescence intensities of the supernatants and the initial BRITC-FBS solution, the adsorption efficiency (%) of FBS was assessed by the following equation:(1)$$\mathrm{Adsorption}\;\mathrm{efficiency}\;\left(\%\right)=(I_{\mathrm{initial}}-I_{\mathrm{supernatant},1})\times 100/I_{\mathrm{initial}}$$
where *I*_initial_ is the fluorescence intensity of the initial BRITC-FBS solution and *I*_supernatant,1_ is the fluorescence intensity of the supernatants at the predetermined time.

These BRITC-FBS-preadsorbed samples were put into fresh PBS (5 mL) and incubated at 37 °C to detect protein retention. At the predetermined time, each sample was centrifuged for 30 s at a speed of 100× *g*. The PE was rinsed with fresh PBS three times to ensure the removal of unadsorbed FBS, and then, the collected supernatants were detected using a fluorescence microscope (A10, Zeiss, Oberkochen, Germany). The fluorescence intensities were recorded for the determination of the protein retention on coatings (Equation (2)).
(2)$$\mathrm{Retention}\;\mathrm{rate}\;\left(\%\right)=(I_{\mathrm{initial}}-I_{\mathrm{supernatant},1}-I_{\mathrm{supernatant},2})\times 100/(I_{\mathrm{initial}}-I_{\mathrm{supernatant},1)}$$
where *I*_initial_ and *I*_supernatant,1_ are the same as Equation (1) and *I*_supernatant,2_ is the fluorescence intensity of the supernatants at the predetermined time point during the protein retention process.

Alkaline phosphatase (ALP) as a model bioactive factor was dissolved in PBS to monitor protein bioactivity. The different coatings were immersed in 1.5 mL of ALP solution (200 μg/mL) at 4 °C for 3 h, followed by the complete rinsing of coatings with fresh PBS. According to the ALP assay kit instruction, these coatings were put into the mixture of ALP reagents 1 and 2 and incubated at 37 °C for 30 min. Then, ALP reagent 3 was added (Appendix A). At 510 nm, the absorbance was confirmed via UV-Vis measurement.

### 2.6. Biological Investigation

#### 2.6.1. Cell Culture

Mouse fibroblast cells (L929 cells) were utilized to study cell response. L929 cells were cultured in DMEM, supplemented with 10% *v*/*v* FBS and 1% penicillin/streptomycin at 37 °C in a humidified 5% CO_2_-containing atmosphere.

#### 2.6.2. Cell Response on Various Coatings

L929 cells were used to study cell response. Herein, cell response mainly involved cell compatibility, adhesion, proliferation, and detachment. The diverse PE plates were sterilized thoroughly via radiation of γ ray and then rinsed with sterile PBS. Afterward, all sterilized PE plates were immersed in 10% FBS-containing DMEM at 37 °C for 5 h. The protein-preadsorbed PE plates were successfully prepared by washing them with PBS three times. L929 cells (24,000 cells/mL) were seeded on the protein-preadsorbed PE plates and cultured at 37 °C in a 5% CO_2_-containing air atmosphere.

After 24 h cultivation, cell compatibility was evaluated by the investigation of viability using a fluorescent live/dead assay. In brief, the cells on the surface of each PE plate were washed twice with PBS, followed by the addition of the staining solution containing Calcein AM and PI and co-incubation at 37 °C for 30 min. The stained cells were washed three times with pre-cooling PBS. The profiles of the cells were scanned by confocal laser scanning microscopy (CLSM, Olympus FV3000, Tokyo, Japan). The CLSM images were analyzed by Image J2 software to confirm cell compatibility, adhesion, and proliferation. In addition, L929 cells were cultured for 48 h, and then the temperature was reduced to 20 °C in a sterile environment to detect the detachment rate of cells on different coatings. The morphology of the cells was observed using optical microscopy (ZEISS, AxioVert5, Oberkochen, Germany) during cell detachment. The number of detached cells in DMEM suspension was recorded by a hemocytometer (Shanghai Qiujing, Shanghai, China) at a predetermined time point.

### 2.7. Statistical Analysis

In this work, every group of experiments was performed in quintuplicate at least. A normality test was employed to estimate data quality, and then, both one-way analysis of variance and Tukey’s multiple comparison tests were used to evaluate the statistical significance of the data. If *p* < 0.05, the data were assumed to be significant. * Represents *p* < 0.05 and ** indicates *p* < 0.01. All data were expressed as the mean value ± standard deviation.

## 3. Results and Discussion

### 3.1. Chemical Components and Content of Various Coatings

The chemical components of the coatings have a considerable effect on interface wettability [26]. To detect the elemental composition of each coating, XPS measurement was implemented; the full spectra of the pristine PE and various PNIPAAm-APTES-TA coatings are shown in Figure 1a. In contrast to the virgin PE with C_1s_ (284.6 eV) and O_1s_ (528.3 eV) peaks, three new peaks appear at 101.2, 164.1, and 400.4 eV for the PNIPAAm-APTES-TA coating-modified PE, which was assigned to Si_2p_, S_2p_, and N_1s_, respectively [27,28,29]. Si_2p_, N_1s_, and S_2p_ were attributed to APTES (N and Si) and PNIPAAm-OH (N and S) (Appendix A), respectively. In this work, all coatings were strictly treated for the removal of residual reagents. In view of the above analysis, PNIPAAm-APTES was successfully synthesized and deposited on PE, as depicted in Figure 1a, b. In addition, Si_2p_ core-level spectra were fitted with two peaks (Appendix A), one of which is located at 101.7 eV, representing the peak of Si-O-C, which further verified the synthesis of PNIPAAm-APTES via the reaction between APTES and PNIPAAm-OH. The oxygen content for PNIPAAm-APTES-TA coatings is higher than the oxygen-containing theoretical value (~14.2%) of the corresponding PNIPAAm-APTES coatings (Figure 1b and Appendix A). The results show that high oxygen-containing TA was introduced in the PNIPAAm-APTES coatings to successfully prepare PNIPAAm-APTES-TA coatings. Moreover, it is worth noting that the content of TA in each PNIPAAm-APTES-TA coating was almost identical due to the spraying of the same amount of TA on every PNIPAAm-APTES coating. However, the oxygen content on the surfaces of the PNIPAAm-APTES-TA coatings has a tendency to increase from PNIPAAm_2000_-APTES-TA to PNIPAAm_500_-APTES-TA (the oxygen content on PNIPAAm_860_-APTES-TA was nearly equal to that of PNIPAAm_500_-APTES-TA). The findings are in relation to different distributions of TA in PNIPAAm-APTES-TA variants. For samples III and IV, TA may prefer to disperse the surfaces of PNIPAAm-APTES-TA, which caused the higher content of oxygen on the surface. Furthermore, TAs in samples I and II may not favor this tendency, and part of them are prone to diffusion into the inner layer of PNIPAAm-APTES-TA, resulting in these TAs occurring beyond the depth of XPS scanning (10 nm) and a lower content of oxygen on the two samples (Figure 1c).

XPS fitting spectra of N_1s_ were used to investigate the relative content of the chemical bonding-TA in coatings. As displayed in Figure 1c, the N_1s_ core-level spectra were fitted with three peaks, corresponding to a secondary amine group (–NH–) at 400.2 eV, a protonated amine group (C–NH_3_^+^) at 401.9 eV, and an aromatic N at 398.9 eV [30]. Notably, the aromatic N and C–NH_3_^+^ stemmed from the reaction of PNIPAAm-APTES with TA by Michael addition (Figure 1a) and the protonated amino group of residual PNIPAAm-APTES, respectively. Furthermore, the relative content of aromatic N in different PNIPAAm-APTES-TA coatings was 13.5% for PNIPAAm_2000_-APTES-TA, 14.7% for PNIPAAm_1300_-APTES-TA, 21.5% for PNIPAAm_860_-APTES-TA, and 28.7% for PNIPAAm_500_-APTES-TA (Figure 1d). The findings indicate that the amount of the covalent bonding-TAs on PNIPAAm_860_-APTES-TA and PNIPAAm_500_-APTES-TA is much more than that of PNIPAAm_2000_-APTES-TA and PNIPAAm_1300_-APTES-TA. In addition, the relative content of nitrogen in PNIPAAm-APTES-TA tends to gradually increase from sample I to IV, as listed in Appendix A. According to the results above, we easily calculated that not all TAs incorporated into PNIPAAm-APTES-TA were covalently bonded to coatings and part of the TAs must be introduced to the coating (PNIPAAm_2000_-APTES-TA and PNIPAAm_1300_-APTES-TA and PNIPAAm_860_-APTES-TA) by a non-chemical bonding effect.

TAs were incorporated into PNIPAAm-APTES-TA by two formations, i.e., chemical bonding and non-chemical bonding, and the content of chemical bonding-TAs gradually enhanced from sample I to IV. According to the distribution of TA on the coatings, PNIPAAm-APTES-TA coatings could be classified into two categories, as depicted in Figure 1c. One type is PNIPAAm_2000_-APTES-TA and PNIPAAm_1300_-APTES-TA, where TAs were prone to dispersion in the entire internal space of PNIPAAm-APTES. Another is PNIPAAm_860_-APTES-TA and PNIPAAm_500_-APTES-TA, in which the TAs were mainly concentrated on the surfaces of PNIPAAm-APTES.

### 3.2. Morphologies of Various Coatings

Interface morphology plays a vital role in tailoring interfacial wetting, further determining cell adhesion and detachment [16]. The shapes of diverse PNIPAAm-APTES hydrogels were investigated by TEM, as shown in Figure 2a. Although all PNIPAAm-APTESs are spherical, the density between these nanospheres seems to be significantly different. PNIPAAm_2000_-APTES and PNIPAAm_1300_-APTES are loose, while PNIPAAm_860_-APTES and PNIPAAm_500_-APTES are relatively dense. After PNIPAAm-APTES was deposited on PE and TA was sputtered, pore-free coatings were observed on the surface of sample I, whereas the coatings on samples II, III, and IV had a porous structure, with an average pore diameter of less than 1 μm (Figure 2b,c). In particular, the spherical nanohydrogels possess a high specific surface area, which is beneficial for accommodating more active sites to bind growth factor/cell adhesion molecules (CAMs) [31]. Additionally, the nanospherical structure could considerably change the surface wetting [32,33]. Nanogels with different densities may lead to the diffusion of TA into them. In dense-structured PNIPAAm_860_-APTES and PNIPAAm_500_-APTES, the diffusion of TA may be difficult, which leads to TAs being blocked on the surface of these nanogels. In contrast, in PNIPAAm_2000_-APTES and PNIPAAm_1300_-APTES with loose structures, TA can easily diffuse into the entire space of these nanogels. Consequently, the distribution of TA for PNIPAAm_860_-APTES-TA and PNIPAAm_500_-APTES-TA is different from that of PNIPAAm_2000_-APTES-TA and PNIPAAm_1300_-APTES-TA (Section 3.1). The findings indicate that PNIPAAm_860_-APTES-TA and PNIPAAm_500_-APTES-TA may provide dramatically different interface microenvironments for cell culture compared to that of PNIPAAm_2000_-APTES-TA and PNIPAAm_1300_-APTES-TA.

### 3.3. Stability of TA-Modified Hydrogel Coatings

UV-Vis measurement was carried out to detect the stability of the TA-modified hydrogel coatings in PBS at pH 7.4. As shown in Figure 3a, the absorbance peak of TA was found in the hydrogel coating-immersed PBS, implying the release of TA for all samples. However, each specimen has an evident distinction in TA release profiles. TA release for PNIPAAm_2000_-APTES-TA was very similar to that of PNIPAAm_1300_-APTES-TA, except for a slightly higher release intensity. Furthermore, similar TA release profiles for PNIPAAm_860_-APTES-TA and PNIPAAm_500_-APTES-TA were also observed, and their release intensities were not significantly different during the first 24 h (Figure 3b). For PNIPAAm_2000_-APTES-TA and PNIPAAm_1300_-APTES-TA, the release rates of TA were over four times that of PNIPAAm_860_-APTES-TA and PNIPAAm_500_-APTES-TA. The results were given in relation to the content of chemical bonding-TA and the distribution of TA in hydrogels. Since TA is a natural polyacid, the deprotonation will occur in the faintly alkaline condition [23]. The hydrogen bond between PNIPAAm-APTES and TA was dramatically impaired and even faded away due to the deprotonation of TA and elevated hydration of the outmost termination of the PNIPAAm chain. The diminished non-chemical interaction of hydrogels with TA easily caused TA release and instability of coating. In addition to more chemical bonding-TAs, TAs in PNIPAAm_860_-APTES-TA and PNIPAAm_500_-APTES-TA were concentrated to populate on the surfaces of hydrogels. The mutual overlaps of aromatic rings of TAs were more likely to occur, creating a cross-linking network via π-π stacking and hydrophobic interactions. The cross-linking networks were firmly fixed on the surfaces of samples III and IV through chemical bonding-TAs. So the TA release was effectively suppressed, and the coatings were relatively stable. In contrast, TAs were dispersed in the entirely internal spaces of PNIPAAm_2000_-APTES-TA and PNIPAAm_1300_-APTES-TA. Consequently, the distance between TAs may be so large that it is difficult to form mutual overlaps of TAs and the stable cross-linking networks of TA may not be created. In addition, fewer TAs were chemically bonded on PNIPAAm_2000_-APTES-TA and PNIPAAm_1300_-APTES-TA, which caused easy TA release. Overall, the TA-modified hydrogel coatings on PNIPAAm_860_-APTES-TA and PNIPAAm_500_-APTES-TAare relatively stable in the long term, which may be better suitable for maintaining the protein on coatings [25,34].

### 3.4. Thermo-Responsiveness of Various Hydrogel Coatings

The thermo-sensitive trait of PNIPAAm is able to promote spontaneous cell detachment from material surfaces by temperature changes, which is important for the enhancement of efficiency for cell sheet culture [29]. To verify the thermo-responsiveness of the coatings, a qualitative analysis of the snatched coating solution was determined using optical images. As depicted in Appendix A, it can be seen that the transparent solution at 20 °C turns into a milky white turbid liquid at 37 °C. The results confirm that all coatings possess thermo-responsiveness and show that the volume phase transition temperatures of hydrogel coatings are in the range of 20–37 °C.

The swelling changes with temperature were utilized to make a quantitative analysis of the thermo-sensitive properties of the coatings. Figure 4 depicts the swelling changes in different PNIPAAm-APTES-TAs with temperature, including a heat cycle, i.e., heating and cooling processes. For these four sorts of hydrogel coatings, when temperature changed in the range of 20–32 °C, the overall swelling tendency was diminishment with heating and elevation with cooling, further demonstrating the thermo-responsiveness of all PNIPAAm-APTES-TA coatings. The results of DLS also exhibited that D_h_ increased with decreasing temperature and decreased with increasing temperature (20 < T < 32 °C), which was similar to the changes in swelling with temperature for PNIPAAm-APTES-TA (Appendix A).The findings may have a close relationship with the changes in wetting with temperature. The hydrophobicity of PNIPAAm-APTES-TA increased with elevating temperature, resulting in the collapse of nanogels, while the hydrophilicity of PNIPAAm-APTES-TA rises with decreasing temperature, which causes swelling of hydrogels. Moreover, there was an evident distinction between PNIPAAm-APTES-TA and PNIPAAm-APTES in terms of swelling rate (Appendix A). For each PNIPAAm-APTES-TA coating, the swelling rate was less than that of corresponding PNIPAAm-APTES coatings, which was related to the limitation of TA on the movement of the PNIPAAm chain. A part of TA in PNIPAAm-APTES-TA coatings served as a cross-linker, which led to a lower swelling ratio compared to that of PNIPAAm-APTES. Additionally, regardless of heating or cooling, the swelling rate from sample I to IV presented a gradual decline. In particular, the swelling from PNIPAAm_1300_-APTES-TA to PNIPAAm_860_-APTES-TA decreased by over 15.1%. The tendency of swelling to decline could be explained by different amounts of TAs as cross-linkers. Based on the analyses of both XPS and coating stability, the TA that acted as a cross-linker or was chemically bonded showed an increasing trend from PNIPAAm_2000_-APTES-TA to PNIPAAm_500_-APTES-TA. These TAs are more abundant, which creates a stronger inhibition of the swelling of coatings. The amount of these TAs in PNIPAAm_860_-APTES-TAshowsan evident increase compared to that of PNIPAAm_1300_-APTES-TA. Thus, the swelling of PNIPAAm_860_-APTES-TA was suppressed more. To a certain extent, TA restricted the movement of the PNIPAAm chain, but these coatings still possess thermo-responsiveness.

The solubility of both PNIPAAm and its copolymer increases with decreasing temperature. These polymers exhibit a LCST. Any diminishment in temperature from the LCST leads to the formation of a single phase, which makes the solution clear. The LCST of each PNIPAAm-APTES-TA coating was detected to further study thermo-responsive behaviors. As shown in Figure 5, a slight reduction in the LCST from PNIPAAm_2000_-APTES-TA to PNIPAAm_500_-APTES-TA is found. The results indicate that the LCST of PNIPAAm-APTES-TA coating decreases with the increase in the incorporated TA that acted as a cross-linker by chemical bonding or non-chemical bonding. These TAs inhibited the transition in the conformation of PNIPAAm-APTES-TA with changes in temperature, which led to the relative difficulty in the formation of a single phase. Generally, further lowering the temperature allowed the PNIPAAm-APTES-TA to obtain enough driving force for the transition in conformation, which can promote the formation of the single-phase solution again. Therefore, the LCST of samples in the range of I–IV exhibited a tendency to decrease.

### 3.5. Wetting of Diverse Hydrogel Coatings

Interface wetting is also a key factor for taking command of protein adsorption as well as subsequent cell adhesion and detachment [35,36,37]. The wetting of pristine PE and PNIPAAm-APTES-TA coatings was estimated by static water contact angle (WCA). As displayed in Figure 6a, the WCAs for all PNIPAAm-APTES-TA coatings dropped dramatically to less than 60° compared to the virgin PE (over 120° at 20 and 37 °C). The findings indicate that these coatings could change interface wetting. In addition, with the incorporation of TA on the PNIPAAm-APTES hydrogels, the WCAs decreased monotonously in the following order: sample I (PNIPAAm_2000_-APTES-TA) > sample II (PNIPAAm_1300_-APTES-TA) > sample III (PNIPAAm_860_-APTES-TA) > sample IV (PNIPAAm_500_-APTES-TA), implying an increase in wettability. Two reasons may explain this phenomenon: (1) the spherical nanostructure of hydrogels that constituted the films of PNIPAAm_2000_-APTES-TAandPNIPAAm_500_-APTES-TA [38,39] and (2) intensive hydrophilic groups (such as an amine group and a phenol hydroxyl group) on the surfaces of PNIPAAm_2000_-APTES-TAandPNIPAAm_500_-APTES-TA [40].

The change in WCA with temperature and time was investigated to ascertain the effect of temperature on the wettability of PNIPAAm-APTES-TA coatings (Figure 6). For each hydrogel coating, the WCA at 20 °C decreased by approximately 20° compared to the corresponding ones at 37 °C.However, the WCA on the virgin PE was hardly influenced by temperature (over 120° at 20 and 37 °C). The findings suggested that the PNIPAAm-APTES-TA coatings displayed relative hydrophobicity at high temperatures and relative hydrophilicity at low temperatures. Hydrophobicity at 37 °C favors protein adsorption such as CAMs and hydrophilicity at 20 °C benefits spontaneous cell detachment. The changes in WCA with time are shown in Figure 6c,d. When the temperature was maintained at 20 or 37 °C, the WCAs on the surfaces of pristine PEs and PNIPAAm_2000_-APTES-TA coatings rarely changed over the long term. For PNIPAAm_1300_-APTES-TA, PNIPAAm_860_-APTES-TA, and PNIPAAm_500_-APTES-TA, although all WCAs were also almost constant at 37 °C, they were quickly reduced to 0° within less than 8 s at 20 °C. The results demonstrate that the inner layers of PNIPAAm_1300_-APTES-TA, PNIPAAm_860_-APTES-TA, and PNIPAAm_500_-APTES-TAcoatings are hydrophilic at 20 °C. In addition, the PNIPAAm_860_-APTES-TA coating and PNIPAAm_500_-APTES-TA coating exhibited superhydrophilicity (WCA < 20°) compared to that of PNIPAAm_1300_-APTES-TA coatings. This was associated with the hydrophilic nanospheres and more TA with hydrophilic groups on the surfaces of PNIPAAm_860_-APTES-TA and PNIPAAm_500_-APTES-TA.

### 3.6. Protein Behaviors on Hydrogel Coatings

Protein adsorption is the first stage of cell adhesion and subsequent cell fate, which plays a dominant role in cell response to the material [41,42]. Thereby, it becomes crucial to determine the protein behaviors on each PNIPAAm-APTES-TA coating. The protein behaviors mainly involved protein adsorption, retention, and bioactivity. Protein adsorptions of PNIPAAm-APTES-TA coatings (sample I–IV) were stronger than that of the corresponding PNIPAAm-APTES coatings (sample V–VIII) (Appendix A), implying that TA indeed conferred the PNIPAAm-APTES coatings with a stronger ability of protein adsorption. Moreover, from PNIPAAm_2000_-APTES-TA coatings to PNIPAAm_500_-APTES-TA coatings (Figure 7a–d), the fluorescence intensity of PNIPAAm-APTES-TA coating-immersed supernatants significantly diminished in the first 5 h, demonstrating that protein was quickly adsorbed onto these coatings. Furthermore, the intensity barely changed in the following time, which demonstrated that protein adsorption reached the equilibrium state. After each PNIPAAm-APTES-TA coating was soaked for 24 h, the intensity of the supernatants decreased by 44.4% for PNIPAAm_2000_-APTES-TA coatings, 53.3% for PNIPAAm_1300_-APTES-TA coatings, 71.1% for PNIPAAm_860_-APTES-TA coatings, and 72.4% for PNIPAAm_500_-APTES-TA coatings, indicating the significantly incremental amount of protein adsorbed on the coatings ranging from PNIPAAm_2000_-APTES-TA coatings to PNIPAAm_500_-APTES-TA coatings. The surfaces of PNIPAAm_860_-APTES-TA coatings and PNIPAAm_500_-APTES-TA coatings had nearly the same amount of adsorbed protein. Based on the XPS results (Section 3.1), the TA content on the surface of PNIPAAm_860_-APTES-TA was almost identical to that of PNIPAAm_500_-APTES-TA. The findings indicated that the amount of protein adsorbed on PNIPAAm-APTES-TA coatings was closely related to the TA content on the surfaces. Since the phenol hydroxyl group and phenyl group of TA can act with protein via hydrogen bonds, π-π stacking, and hydrophobicity [34], with the incorporation of more TA on the surface, PNIPAAm_860_-APTES-TA and PNIPAAm_500_-APTES-TA exhibited greater protein adsorption.

Owing to the continuous supply of protein being essential for the processes in the cell, the protein retention capability on the coatings is also vital for cell culture. The protein retention rate is displayed in Figure 8b and Appendix A. The protein retention rate of CPE is less than 10%. For the control, i.e., PNIPAAm-APTES hydrogel coatings, the protein retention rate from PNIPAAm_2000_-APTES to PNIPAAm_500_-APTES was around 11.7%. The results showed that the PNIPAAm-APTES hydrogels were not able to increase protein retention in comparison to PE. However, the retention rate on the diversity of PNIPAAm-APTES-TA reached more than 45%, which implied their outstanding protein retention capability. The dominant reason for the high protein retention may be π-π stacking and hydrophobic interactions of TA with hydrophobic amino acid residues of proteins [43,44]. The retention rate of protein on PNIPAAm-APTES coatings (sample V–VIII) decreased monotonously with time during the entire protein retention period, which was consistent with the report by Nagahama [45]. The protein retention rates of PNIPAAm_2000_-APTES-TA and PNIPAAm_1300_-APTES-TA in the first 12 h were 37.5% and 38.9%, respectively. After they were immersed for 24 h, the protein retention rates increased to 48.3% and 50.6%, respectively. This abnormal result was ascribed to the deposition of the complex of the released TA with protein. During the whole immersion term (24 h), the retention of protein on PNIPAAm_860_-APTES-TA and PNIPAAm_500_-APTES-TA declined slowly (Appendix A), suggesting that the stable coatings could effectively modify protein retention. Moreover, Figure 8a shows the protein distribution profiles on every sample before and after the immersion of protein-preadsorbed coatings in PBS for 24 h. Despite the fact that protein distribution on every sample was relatively homogeneous before the coatings were soaked, a huge change took place after the coatings were immersed for 24 h. For PNIPAAm_2000_-APTES-TA and PNIPAAm_1300_-APTES-TA, the distribution of protein became heterogeneous, and the aggregation of protein appeared on the surfaces. However, after the protein-preadsorbed PNIPAAm_860_-APTES-TA/PNIPAAm_500_-APTES-TA was soaked in PBS for 24 h, although the amount of protein decreased, the distribution of protein exhibited relatively uniform profiles. The results implied that the distribution of protein was also affected by the stability of the coating. The stable coating on PNIPAAm_860_-APTES-TA and PNIPAAm_500_-APTES-TA could maintain uniform protein distribution during protein retention.

Given that strong electrostatic and/or hydrophobic interactions between the material interface and the protein reduce the bioactivity of the adsorbed protein [46,47,48], the protein bioactivity was evaluated using a protein model. ALP served as a model protein to estimate the bioactivity of the adsorbed protein on the PNIPAAm-APTES-TA coatings. Appendix A and Figure 8c show the results of the bioactivity of adsorbed ALP on the different coatings. After the ALP reagents were treated by each ALP-immersed PNIPAAm-APTES-TA coating, the color of the solution changed from colorless to red, and the absorbance at 510 nm was detected. However, after treatment with the corresponding ALP-free coatings, the ALP reagent was still colorless and had no absorbance at 510 nm. The findings indicate that ALP was not only adsorbed on PNIPAAm-APTES-TA coatings but also expressed good bioactivity, and TA incorporated into coatings has no considerable negative effect on the protein bioactivity of PNIPAAm-APTES-TA coatings.

### 3.7. Cell Response on Diverse Coatings

Cell adhesion as the original response of the cell to a foreign surface has a profound impact on the cell fate on a material’s surface [5,6]. Adhesion area and aspect ratio were utilized to characterize the cell adhesion behaviors on diverse samples [49]. The CLSM images of cells were quantified by Image J, as shown in Figure 9c,d. The adhesion area and aspect ratio of cells on PNIPAAm_860_-APTES-TA and PNIPAAm_500_-APTES-TA were 965–1010 μm^2^ and 7.4–7.6, respectively, whereas the corresponding ones on PNIPAAm_2000_-APTES-TA and PNIPAAm_1300_-APTES-TA were 486–630 μm^2^ and 2.1–3.5, respectively. The adhesion area and aspect ratio of cells on PNIPAAm_860_-APTES-TA and PNIPAAm_500_-APTES-TA were similar to those of CPE (1100 μm^2^ and 7.1), which has been commercially applied, and are larger than the corresponding values of PNIPAAm_2000_-APTES-TA and PNIPAAm_1300_-APTES-TA. The findings suggest that the stable TA-modified PNIPAAm hydrogel coatings are beneficial to enhance cell adhesion. Cell viability was investigated by CLSM images of live/dead staining cells. The viability and distribution of cells on various coatings are evidently different (Figure 9a,b and Appendix A). For PNIPAAm_2000_-APTES-TA and PNIPAAm_1300_-APTES-TA, fewer live cells were observed, and live cells were preferable to aggregate together. By contrast, live cells were nearly 100% and intensively populated with higher cell numbers on PNIPAAm_860_-APTES-TA and PNIPAAm_500_-APTES-TA. Although the aggregation of cells was not observed on CPE, the viability was the lowest, with only around 50%. Additionally, the growth of cells on PNIPAAm_860_-APTES-TA and PNIPAAm_500_-APTES-TA also displayed more expansion (Appendix A). The findings manifest that PNIPAAm_860_-APTES-TA and PNIPAAm_500_-APTES-TA provide a more suitable environment of cell culture compared to PNIPAAm_2000_-APTES-TA and PNIPAAm_1300_-APTES-TA.

The spontaneous detachment of cells is vital to improve the viability of the cell culture process [7]. The morphology and detachment rate on the various surfaces of PE plates were obtained through in situ observation using optical microscopy and real-time record of the detached cell number using a hemocytometer, respectively. Figure 10a–d exhibits the changes in the morphology of the cells on different PNIPAAm-APTES-TA coatings after the predetermined incubation time of the cells at 20 °C. From 0 to 9 min, when the temperature decreased to 20 °C, part of the cells on each PNIPAAm-APTES-TA coating displayed an evident change in morphology, i.e., from the spindle/elongated shape toward a circular one. A similar change was also found by Raczkowska [50]. For the bare surface of CPE, no changes in cell morphology were found after lowering the temperature to 20 °C (Figure 10e). The results imply that lowering the temperature to below the LCST can drive cell detachment. Moreover, we found that the initial cells on PNIPAAm_860_-APTES-TA and PNIPAAm_500_-APTES-TA were sufficiently stretched, whereas the ones on PNIPAAm_2000_-APTES-TA and PNIPAAm_1300_-APTES-TA presented an incompletely stretched shape. The difference in initial state between these cells may relate to the diverse protein behaviors on coatings, which is in line with the conclusion of CLSM. The change in the morphology of the cells on PNIPAAm_860_-APTES-TA and PNIPAAm_500_-APTES-TA (from a sufficiently stretched shape to a round one) is greater than that on PNIPAAm_2000_-APTES-TA and PNIPAAm_1300_-APTES-TA (from the incompletely stretched shape to around one) under cell incubation at 20 °C. The findings indicate that lowering the temperature to below the LCST has a greater influence on the change in the morphology of the cells on PNIPAAm_860_-APTES-TA and PNIPAAm_500_-APTES-TA compared to that on PNIPAAm_2000_-APTES-TA and PNIPAAm_1300_-APTES-TA. Figure 9e exhibits the detachment rate of cells on various surfaces with time. When the temperature was reduced to 20 °C, in less than 10 min, the detachment rates of cells on PNIPAAm_860_-APTES-TA and PNIPAAm_500_-APTES-TA separately reached 85.4% and 92.2%, while the ones on PNIPAAm_2000_-APTES-TA and PNIPAAm_1300_-APTES-TA were about 78.1% and 67.6%, respectively. As a control, cell detachment cannot occur spontaneously on CPE. The results further confirmed that the PNIPAAm-APTES-TA hydrogels were able to promote cell detachment at low temperatures. In particular, the coatings on PNIPAAm_860_-APTES-TA and PNIPAAm_500_-APTES-TA possessed better properties, with cell adhesion at 37 °C and detachment at 20 °C.

Responses of cells to the interface of foreign materials may be impacted by multiple factors such as softness, wetting, morphologies, and stability of the coating. Since PNIPAAm is a thermo-sensitive polymer, the softness of its hydrogels relies mainly on temperature [51]. Thus, the softness of hydrogel coatings is unable to affect cell adhesion, viability, and distribution. In general, the hydrophobic surfaces favor protein adsorption and cell attachment [16]. However, better cell adhesion was exhibited on more hydrophilic PNIPAAm_860_-APTES-TA and PNIPAAm_500_-APTES-TA compared to that of PNIPAAm_2000_-APTES-TA and PNIPAAm_1300_-APTES-TA. The results denoted that wettability was also not the predominant factor affecting cell response. Spherical nanohydrogels on PNIPAAm_860_-APTES-TA and PNIPAAm_500_-APTES-TA possess higher specific surface area compared to PNIPAAm_2000_-APTES-TA and PNIPAAm_1300_-APTES-TA, which can provide more active sites for CAMs and subsequent cell adhesion. Thereby, the interface morphology may be a factor that affects cell response. Since the PNIPAAm-APTES-TA coating depended mainly on TA to regulate the protein behaviors on their surface such as the adsorption or desorption of CAMs, CAMs that are involved in cell adhesion could be lost with TA release. Consequently, the stabilities of different PNIPAAm-APTES-TA coating variants may have a profound influence on cell response. The cross-linking TA networks on PNIPAAm_860_-APTES-TA and PNIPAAm_500_-APTES-TA nanohydrogels formed via the synergistic effect of chemical bonding with non-chemical bonding can improve the stability of the coatings. These stable PNIPAAm-APTES-TA coatings effectively restricted TA release. So, the CAM could be stably adsorbed on these coatings, thereby promoting cell adhesion. On the contrary, TAs were readily released from the unstable PNIPAAm_2000_-APTES-TA and PNIPAAm_1300_-APTES-TA coatings, which blocked the adsorption of CAMs and subsequently, cell adhesion and viability. Furthermore, under the attraction of the released TA, CAMs desorbed from PNIPAAm_2000_-APTES-TA and PNIPAAm_1300_-APTES-TA could be gathered together to form the CAM’s aggregations. These aggregations are redeposited on the local regions of the surfaces rather than being uniformly dispersed on the whole surfaces, resulting in the movement of more cells to CAM-enriched regions and subsequent cell aggregation. In a word, the stability of the PNIPAAm-APTES-TA coatings influences not only the cell adhesion and viability but also the cell distribution; a better response of cells on PNIPAAm_860_-APTES-TA and PNIPAAm_500_-APTES-TA with the cross-linking TA networks was expressed in comparison with that on PNIPAAm_2000_-APTES-TA and PNIPAAm_1300_-APTES-TA.

Spontaneous cell detachment for various hydrogel coatings arose from the change in wetting of thermo-responsive PNIPAAm. In comparison with CPE, a significant cell detachment was found for each coating after the temperature dropped to 20 °C for 10 min. This result may be explained by the better hydrophilicity of PNIPAAm at low temperatures. Moreover, cell detachment on PNIPAAm_860_-APTES-TA and PNIPAAm_500_-APTES-TA took less time than that on PNIPAAm_2000_-APTES-TA and PNIPAAm_1300_-APTES-TA, which was attributed to the superhydrophilic surfaces [16,52]. It was the synergistic effect between spherical hydrogels and hydrophilic groups of TA that endowed both PNIPAAm_860_-APTES-TA and PNIPAAm_500_-APTES-TA with superhydrophilicity. Thus, taking all the above factors into consideration, the response of cells on TA-modified PNIPAAm hydrogels can be modulated by the morphologies and stability of the hydrogel coatings. In particular, the efficiency of cell culture can be enhanced by an improvement in cell adhesion and detachment via the modulation of the stability of PNIPAAm coating from chemical bonding-TA.

## 4. Conclusions

Amino-terminated PNIPAAm (PNIPAAm-APTES) containing different amine group contents were successfully synthesized and spontaneously deposited on TA/Fe^3+^-modified PE plates. Subsequently, TA was incorporated into the previously mentioned PNIPAAm-APTES hydrogels to prepare TA-modified PNIPAAm (PNIPAAm-APTES-TA) hydrogels, each with varying chemically bonded TA contents, designated as PNIPAAm_2000_-APTES-TA, PNIPAAm_1300_-APTES-TA, PNIPAAm_860_-APTES-TA, and PNIPAAm_500_-APTES-TA, for cell culture applications. Characterizations using TEM and SEM confirmed the porous structures of coatings composed of spherical PNIPAAm_860_-APTES-TA and PNIPAAm_500_-APTES-TA. For PNIPAAm_860_-APTES-TA and PNIPAAm_500_-APTES-TA, the results of XPS demonstrated that more TAs were left on the surfaces and covalently bonded onto these hydrogels in comparison with PNIPAAm_2000_-APTES-TA and PNIPAAm_1300_-APTES-TA. This contributed to the creation of a more stable hydrogel coating. In particular, PNIPAAm_860_-APTES-TA and PNIPAAm_500_-APTES-TA could improve protein adsorption, prolong protein retention, and ensure protein viability by the suppression of TA release. Cells cultured onPNIPAAm_860_-APTES-TA and PNIPAAm_500_-APTES-TA possessed better cell adhesion, viability, and proliferation compared to that of PNIPAAm_2000_-APTES-TA and PNIPAAm_1300_-APTES-TA. Furthermore, cells cultured on PNIPAAm_860_-APTES-TA and PNIPAAm_500_-APTES-TA spontaneously detached in less than 10 min at 20 °C, attributable to the interfacial superhydrophilicity, thereby demonstrating higher efficiency in cell culture applications.

## Data Availability

The data can be shared up on request.

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
