# Peer review of "In Situ Preparation of Tannic Acid-Modified Poly(N-isopropylacrylamide) Hydrogel Coatings for Boosting Cell Response"

_pharmaceutics, 2024, doi:10.3390/pharmaceutics16040538_

Round 1

Reviewer 1 Report

Comments and Suggestions for Authors

Authors
The authors present an article that may be of interest to the readership of Pharmaceutics and in my opinion can eventually be accepted after some revisions.
I leave some points to the consideration of the authors:
page 2-3, section 2.1. Materials – Purity of the purchased reagents should be stated
p. 3, section 2.2, line 7 – AT – Do the authors mean TA?
p. 16, line 4 – ATPES (2 instances) – Do the authors mean APTES?

I wish the authors all the best in the continuing of their work.

Comments on the Quality of English Language

page 1: Please check lines 4/5 of Abstract: spon taneously

p 2, lines 17-18: the extra process - Please consider something like: the extra processes

p 2, paragraph 2, line 10: two folds - Please consider something like: two times the
p. 8, line 8 from end: thecoatings
p. 14 line 17 – founds - Please consider something like: findings

I wish the authors all the best in the continuing of their work.

Author Response

Reviewer #1 The authors present an article that may be of interest to the readership of Pharmaceutics and in my opinion can eventually be accepted after some revisions.

    Author’s reply: We highly appreciate the reviewer’s time and constructive advices on our manuscript. The reviewer also pointed out imperfections and some contents that need to be improved in the manuscript, and give our some constructive suggestions. We take these comments seriously and make carefully revisions to improve the quality of the manuscript. Below we will reply to the reviewer’s questions and comments point by point in the scope of the present revised manuscript.

1. Page 2-3, section 2.1. Materials – Purity of the purchased reagents should be stated.

    Author’s reply: We think that this advice is very constructive for the improvement of the manuscript and the purity of the related reagents had been added in Section 2.1 of the revised manuscript. The all changes have been highlighted using the yellow background.

2. Page 3, section 2.2, line 7 – AT – Do the authors mean TA? Page 16, line 4 – ATPES (2 instances) – Do the authors mean APTES? Page 1: Please check lines 4/5 of Abstract: spontaneously

    Author’s reply: We are very sorry for these mistakes. In the original manuscript, the TA and APTES were miswritten to AT (Section 2.2, line 7) and ATPES (Page 16, line 4), respectively. These mistakes have been corrected carefully in the present revised version. The corrected TA and APTES separately are include in Section 2.2, and Conclusions, and marked by the yellow background. In addition, the blank between line 4 and 5 in Page 1 were also deleted.

3. Page 2, lines 17-18: the extra process - Please consider something like: the extra processes. Page 2, paragraph 2, line 10: two folds - Please consider something like: two times. The Page 8, line 8 from end: the coatings. Page 14 line 17 – founds - Please consider something like: findings

    Author’s reply: We highly appreciate the reviewer’s valuable advices. According to the review’s advice, the corresponding changes are made such as substitution extra process with extra processes (Page 2, lines 17-18), two folds replaced by two times (Page 2, paragraph 2, line 10). The “founds” are also altered by “findings”.

Reviewer 2 Report

Comments and Suggestions for Authors

The paper titled "In situ preparation of tannic acid-modified poly(N-isopropylacrylamide) hydrogel coatings for boosting cell response" has rich experimental materials, including interesting and new ideas. However, the structure of the paper should be improved before it can be accepted for publication. 

It will be helpful to provide information on smart polymers in the introduction, as some readers may not be familiar with this subject. I propose incorporating this information soon using novel reviews, such as https://doi.org/10.1002/tcr.202300217

I believe the results should start with the chemical structure of the hydrogel, since I suggest transferring subsection "3.2. Chemical components and content of various coatings" in the first place. 

Scheme 1 is hard to understand; chemical reactions from supporting information should be included in Scheme 1. In addition, a detailed explanation of the scheme should be added to the text.

What is the ratio between tannic acid, poly(N-isopropylacrylamide), and APTES? Maybe it makes sense to conduct TG analysis to characterize hydrogel coatings in a more complex manner.

What about cellular morphologies, which suggest positive or negative conditions for cells on the coating? The appropriate discussion should be added. Perhaps it would be useful to compare the cell growth and morphology results with those from T. Okano's group or https://doi.org/10.1021/acsbiomaterials.3c00917

In Fig. 4, the different swelling rates at cooling and heating were shown; this fact was omitted in the text and should be explained. Additionally, the character of curves, which are depicted in Fig. S4, is completely unclear to me. Unfortunately, their explanation is absent in the main text. What are the LCSTs for the studied samples? I expected to see a decrease in the LCST of PNIPAM under the impact of the TA.

Comments on the Quality of English Language

Minor editing of English language required.

Author Response

Reviewer #2 The paper titled "In situ preparation of tannic acid-modified poly(N-isopropylacrylamide) hydrogel coatings for boosting cell response" has rich experimental materials, including interesting and new ideas. However, the structure of the paper should be improved before it can be accepted for publication.

    Author’s reply: We highly appreciate the reviewer’s time and constructive advices on our manuscript. The reviewer also pointed out some contents that need to be further improved in the manuscript and give our some constructive suggestions. We take these comments seriously and make major revisions to improve the quality of the manuscript. Below we will reply to the reviewer’s questions and comments point by point in the scope of the present revised manuscript

1.It will be helpful to provide information on smart polymers in the introduction, as some readers may not be familiar with this subject. I propose incorporating this information soon using novel reviews, such as https://doi.org/10.1002/tcr.202300217

    Author’s reply: We fully agree with the reviewer’s advice, think the recommended paper is very relevant to improve our manuscript, and insert it as Ref [19], to the introduction of the revised manuscript.

2. I believe the results should start with the chemical structure of the hydrogel, since I suggest transferring subsection "3.2. Chemical components and content of various coatings" in the first place.

    Author’s reply: We agree with the reviewer’s opinion that changes the position between chemical components and topographies. This is beneficial to improvement of the quality of manuscript. In the revised manuscript, we rearrange the order of chemical components and topographies of coatings, and some texts also are make some change in response to the adjustment of order. All changes have been marked using yellow background.

3. Scheme 1 is hard to understand; chemical reactions from supporting information should be included in Scheme 1. In addition, a detailed explanation of the scheme should be added to the text.

    Author’s reply: We agree with the reviewer on this point. According to reviewer’s advices, the Scheme 1 is redrawn and is further explained. All changes in Section 3.1 and 3.2 in revised manuscript are highlighted using yellow background.

4. What is the ratio between tannic acid, poly(N-isopropylacrylamide), and APTES? Maybe it makes sense to conduct TG analysis to characterize hydrogel coatings in a more complex manner.

    Author’s reply: The ratio between tannic acid (TA) and amino-terminated PNIPAAm (PNIPAAm-APTES) was listed in Table 1 of original manuscript. The ratio between APTES and hydroxyl-terminated PNIPAAm (PNIPAAm-OH) was also listed in Table S1. Because the amount of N-isopropylacrylamide (PNIPAAm) was more than that of that of hydroxyl-containing 2-mercaptoethanol (2-ME) in synthesis of PNIPAAm-OH, the amount of PNIPAAm-OH can be regarded as that of PNIPAAm. Thereby, the ratio of TA, PNIPAAm and APTES had been listed in Table 1 and S1.

Fig. R1 TGA curves of PNIPAAm2000-APTES-TA (i), PNIPAAm1300-APTES-TA (ii), PNIPAAm860-APTES-TA (iii) and PNIPAAm500-APTES-TA (iv).

    TG analysis is performed to characterize hydrogel coatings as shown the Fig R1. The decomposition of various PNIPAAm-APTES-TA is in the range of 250–420°C. No evident change in the thermostability between different PNIPAAm-APTES-TA was found. Thus, we don’t insert the content of TG into the revised text. In our study, hydrogels coatings are mainly used to culture cells, and liquid environment of cell culture is the application environment of these coatings. The temperature that the coatings were used is not higher than 40°C. Herein, at below 37°C, the thermoresponsive behavior of coatings has been study via UV, DLS and swelling measurement.

5. What about cellular morphologies, which suggest positive or negative conditions for cells on the coating? The appropriate discussion should be added. Perhaps it would be useful to compare the cell growth and morphology results with those from T. Okano's group or https://doi.org/10.1021/acsbiomaterials.3c00917

    Author’s reply: We observed the change of morphology of cells on various coatings when lowering temperature to 20°C using optical microscopy. Fig. 10 was drawn and inserted into the revised manuscript. We explain the change of cell morphology and made a discussion in paragraph 2 of Section 3.7. We got the similar phenomenon with the recommended reference. The results of the recommended study were used to compare with our work. This reference was also cited in our work as Ref. [52]. The content to depict the change of cell morphology with dropping temperature to below LCST was highlighted by yellow background (paragraph 2 of Section 3.7).

6. In Fig. 4, the different swelling rates at cooling and heating were shown; this fact was omitted in the text and should be explained. Additionally, the character of curves, which are depicted in Fig. S4, is completely unclear to me. Unfortunately, their explanation is absent in the main text. What are the LCSTs for the studied samples? I expected to see a decrease in the LCST of PNIPAM under the impact of the TA.

    Author’s reply: We have explained for the different swelling rates during cooling and heating in the Paragraph 2 of Section 3.4 of the revised manuscript and mark the related content by yellow background. We detect the hydrodynamic diameter (Dh) of both PNIPAAm2000-APTES-TA and PNIPAAm1300-APTES-TA again using Dynamic light scanning (DLS) measurement. The results are similar to the Fig. S4 a and b in the original manuscript. We think the fact that Dh of both PNIPAAm2000-APTES-TA and PNIPAAm1300-APTES-TA increase with heating, may result from the TA release. TA in PNIPAAm2000-APTES-TA and PNIPAAm1300-APTES-TA was verified to be more easily release compared with that of PNIPAAm860-APTES-TA and PNIPAAm500-APTES-TA in Section 3.3. In the heating process, for the PNIPAAm2000-APTES-TA and PNIPAAm1300-APTES-TA, TA was rapidly released into solution, the hydrogen-bonding effect that maintained the inner weak cross-linking network of nanogels was impaired, resulting in the fact that the swelling of nanogels overcame the collapse that originated from the increasing hydrophobicity of polymer chains. Thus, the curves of Dh of both PNIPAAm2000-APTES-TA and PNIPAAm1300-APTES-TA with heating look abnormal. In contrast, the TA in PNIPAAm860-APTES-TA and PNIPAAm500-APTES-TA is relative difficult to be released. So the impact of TA's release on cross-linking network is negligible. In the heating process, only the fact that the increasing hydrophobicity of polymer chains caused the collapse of nanoparticle, was displayed. So the curves in which the Dh decreases with elevating temperature were presented for PNIPAAm860-APTES-TA and PNIPAAm500-APTES-TA.

Figure S4 (the original Supporting information) Alternation of hydrodynamic radius with temperature: PNIPAAm2000-APTES-TA (a), PNIPAAm1300-APTES-TA (b), PNIPAAm860-APTES-TA (c) and PNIPAAm500-APTES-TA (d). (the first heat cycle).

    Additionally, we also make the second heat cycle for the samples I, II, III and IV. For the sample III (PNIPAAm860-APTES-TA) and IV (PNIPAAm500-APTES-TA), the curves of Dh with temperature are agreement with the first ones. But the curves of Dh with temperature for sample I (PNIPAAm2000-APTES-TA) and II (PNIPAAm1300-APTES-TA) turn into the ones of typical thermoresponsive behavior (Fig. S4 in the revised Supporting information). The results demonstrate that a new equilibrium between the released TA and nanogels is developed through suppression of the second TA release by the released TA in PBS during the first heat cycle. Thereby, in order to avoid the author’s confusion, we use the data of the second heat cycle to replace the first. The new Fig.S4 was drawn in the revised supporting information.

Figure S4 (the revised Supporting information) Alternation of hydrodynamic radius with temperature: PNIPAAm2000-APTES-TA (a), PNIPAAm1300-APTES-TA (b), PNIPAAm860-APTES-TA (c) and PNIPAAm500-APTES-TA (d). (the second heat cycle)

    In a word, according to the results of our further study, the fact that the swelling of PNIPAAm2000-APTES-TA and PNIPAAm1300-APTES-TA with more easily TA release overcame the collapse of nanogels that resulted from the increasing hydrophobicity with the increase temperature was able to explain the abnormal curves in Fig. S4a and b of the original Supporting information. We also describe the condition of more detailed swelling measurement in Section 2.4 of the revised manuscript. The LCST ranging from sample I to IV is 30.4, 29.3, 27.7 and 26.9°C, respectively, as shown in the Fig. 5 of the revised manuscript and the discussion about Fig. 5 was shown in the Paragraph 3 of Section 3.4 of the revised manuscript

Fig. 5 LCST behavior of each sample. PNIPAAm2000-APTES-TA (i),  PNIPAAm1300-APTES-TA (ii), PNIPAAm860-APTES-TA (iii) and PNIPAAm500-APTES-TA (iv).

Reviewer 3 Report

Comments and Suggestions for Authors

Doctors Hailong Yuan and Tianyou Yang and their colleagues submitted for review a manuscript entitled "In situ preparation of tannic acid-modified poly(N-isopropylacrylamide) hydrogel coatings for boosting cell response”.

The authors developed facile approach to prepare bioactive poly(N-isopropylacrylamide) (PNIPAAm) for cell culture applied in the cellular therapies and tissue engineering. The authors successfully deposited these synthesized nanohydrogels on polyethylene (PE) plates modified with tannic acid (TA) to produce nanohydrogels. The produced nanohydrogels were fully characterized by TEM, SEM, XPS and UV-vis spectroscopy. This manuscript is suitable for rapid publication in the journal Pharmaceutics after discussing the state of the art regarding the synthesis of intermediates, including the mechanism of the reactions taking place, and correcting several errors listed below:

The Abbreviations section is necessary because the manuscript gives the impression of being encrypted due to the necessary multi-letter abbreviations of the names of substances and phenomena that often appear next to each other in the text. This element will contribute to a better understanding of the content of the manuscript by MDPI readers, specialists in various related fields.

At first line of the abstract is: … poly(N-iso … , but should be … poly(N-iso … . Comment: According to the IUPAC nomenclature, it is recommended to write heteroatom symbols in italics in names. See, for example, the title and keywords. Please check and revise in the body of the manuscript.

In the abstract is written … 20oC … , but should be … 20°C … . For example, in the introduction (page 2) it is written correctly ... 37°C … . Similar mistaken is at pages 3, 5, 10, 11 (five times), 14 (twice) and 16 to corrections.  

The introduction says ... Therefore,TA ... , but it should be ... Therefore, TA ... . Comment: Please insert a space character after the comma.

In 2.2 there is ... dehydration condensation reaction ..., maybe it would be better to be descriptive ... condensation reaction with the elimination of water ... .

In 3.3 there is ... coatings[24,33]. ... , but it should be ... coatings [24,33]. ... . Comment, please add a space before the bracket.

  Nowadays, technical journals use the middle sign "–" between numbers, but authors still use the old short sign "-". Please make thorough corrections in the manuscript, both in the references in the main part and in the pages of the source literature cited.   For example, in literature 15 the article number should be ... 6365 ..., and not the page range in the printed version. Please check all source literature and correct if necessary. For example, please correct reference number 23.

In the case of Scheme S1 (a), the authors forgot to refer to the state of the art, for example see DOI: 10.1246/cl.2006.282 of “Reversible Temperature-dependent Dispersion–Aggregation Transition of Poly(N-isopropylacrylamide)–[60]Fullerene Conjugates” by Yajima et all Chemistry Letters, 2006, 35(3), 282–283, and please check similarly and refer to the state of knowledge in the entire experimental part.

Please provide the purity and manufacturer of carbon dioxide, as the repeatability of experimental results may depend on the quality of this basic compound.

Author Response

Reviewer #3 Doctors Hailong Yuan and Tianyou Yang and their colleagues submitted for review a manuscript entitled "In situ preparation of tannic acid-modified poly(N-isopropylacrylamide) hydrogel coatings for boosting cell response”. The authors developed facile approach to prepare bioactive poly(N-isopropylacrylamide) (PNIPAAm) for cell culture applied in the cellular therapies and tissue engineering. The authors successfully deposited these synthesized nanohydrogels on polyethylene (PE) plates modified with tannic acid (TA) to produce nanohydrogels. The produced nanohydrogels were fully characterized by TEM, SEM, XPS and UV-vis spectroscopy. This manuscript is suitable for rapid publication in the journal Pharmaceutics after discussing the state of the art regarding the synthesis of intermediates, including the mechanism of the reactions taking place, and correcting several errors listed below:

    Author’s reply: We highly appreciate the reviewer’s time and constructive advices on our manuscript. The reviewer also pointed out imperfections and some contents that need to be improved in the manuscript. We take these comments seriously and make major revisions to improve the quality of the manuscript. Below we will reply to the reviewer’s questions and comments point by point in the scope of the present revised manuscript.

1. The Abbreviations section is necessary because the manuscript gives the impression of being encrypted due to the necessary multi-letter abbreviations of the names of substances and phenomena that often appear next to each other in the text. This element will contribute to a better understanding of the content of the manuscript by MDPI readers, specialists in various related fields.

    Author’s reply: We fully agree with the reviewer on this point. The acronyms of the names of substances have been used to replace the most sample code in the revised manuscript.

2. At first line of the abstract is: … poly(N-iso … , but should be … poly(N-iso … . Comment: According to the IUPAC nomenclature, it is recommended to write heteroatom symbols in italics in names. See, for example, the title and keywords. Please check and revise in the body of the manuscript.

    Author’s reply: This advice is very helpful for the enhancement of standardability of manuscript. We have corrected it in the updated version.

3. In the abstract is written … 20oC … , but should be … 20°C … . For example, in the introduction (page 2) it is written correctly ... 37°C … . Similar mistaken is at pages 3, 5, 10, 11 (five times), 14 (twice) and 16 to corrections. The introduction says ... Therefore, TA ... , but it should be ... Therefore, TA ... . Comment: Please insert a space character after the comma. In 3.3 there is ... coatings[24,33]. ... , but it should be ... coatings [24,33]. ... . Comment, please add a space before the bracket.

    Author’s reply: We are really sorry about these. We have corrected problems or re-edited the related content thoroughly in the revised manuscript.

4. In 2.2 there is ... dehydration condensation reaction ..., maybe it would be better to be descriptive ... condensation reaction with the elimination of water ... .

    Author’s reply: We highly appreciate the reviewer’s constructive advices about the express of condensation reaction. The corrected expresses are shown in Section 2.2, line 1-3 of the revised manuscript.

5. Nowadays, technical journals use the middle sign "–" between numbers, but authors still use the old short sign "-". Please make thorough corrections in the manuscript, both in the references in the main part and in the pages of the source literature cited.   For example, in literature 15 the article number should be ... 6365 ..., and not the page range in the printed version. Please check all source literature and correct if necessary. For example, please correct reference number 23.

    Author’s reply: We are very sorry for our mistakes about the application of middle sign"-" both in the references in the main part and in the pages of the source literature cited. We have made thoroughly corrections in the revised manuscript and the related corrections have been marked by yellow background in References.

6. In the case of Scheme S1 (a), the authors forgot to refer to the state of the art, for example see DOI: 10.1246/cl.2006.282 of “Reversible Temperature-dependent Dispersion–Aggregation Transition of Poly(N-isopropylacrylamide)–[60]Fullerene Conjugates” by Yajima et all Chemistry Letters, 2006, 35(3), 282–283, and please check similarly and refer to the state of knowledge in the entire experimental part.

     Author’s reply: We would like to thank the reviewer’s constructive suggestions and supply us a reference that is so relevant our study. We cited this paper and insert it into the revised Supporting information as Ref. [2].

7. Please provide the purity and manufacturer of carbon dioxide, as the repeatability of experimental results may depend on the quality of this basic compound.

    Author’s reply: The purity and manufacturer of carbon dioxide used for cell culture in this work is 99.999% and Guangzhou spectral source gas Co., Ltd, China, respectively. In the Section 2.1 of the revised manuscript, we not only have made an introduction for carbon dioxide and but also supply the purity and manufacturer of other reagents that were used in our study

Reviewer 4 Report

Comments and Suggestions for Authors

Authors presented fabricated TA modified PNIPAAm-APTES coatings with superhydrophilicity at 20oC and thermo-responsive behavior. They present that the coatings enhanced cell adhesion and detachment, facilitating efficient cell culture. I think that the manuscript could be published after revision.

 Therefore, I have some questions/suggestions for the authors:

1) How exactly were the AT/Fe3+ tannic acid-modified polyethylene (PE) plates fabricated? Please provide experimental method on this and explain the term “plate”. Why are they called plates? Maybe an additional scheme and optical photographs of the PE plates would be helpful.

2) What reagent is PI (containing Calcein AM and PI) mentioned in section 2.6.2?

3) How do the authors explain the hysteresis on the alternation of hydrodynamic radius observed during heating and cooling step, for samples PE-I (a), PE-II (b) in Figure S4?

4) Τhe English language should be polished. For example, throughout the main text as well as the supplementary material, words and abbreviations are mistaken (e.g. AT/Fe3+ modified PEs, Iinitail, PNIPNNm, PNIPAAm-APEST, SEM imagines).

Comments on the Quality of English Language

Τhe English language should be polished. For example, throughout the main text as well as the supplementary material, words and abbreviations are mistaken (e.g. AT/Fe3+ modified PEs, Iinitail, PNIPNNm, PNIPAAm-APEST, SEM imagines).

Author Response

Reviewer #4 Authors presented fabricated TA modified PNIPAAm-APTES coatings with superhydrophilicity at 20oC and thermo-responsive behavior. They present that the coatings enhanced cell adhesion and detachment, facilitating efficient cell culture. I think that the manuscript could be published after revision.

    Author’s reply: We highly appreciate the reviewer’s time and constructive advices on our manuscript. The reviewer also pointed out imperfections in the original manuscript and gave our some constructive suggestions. We take these comments seriously and make major revisions to improve the quality of the manuscript. Below we will reply to the reviewer’s questions and comments point by point in the scope of the present revised manuscript.

1. How exactly were the TA/Fe3+ tannic acid-modified polyethylene (PE) plates fabricated? Please provide experimental method on this and explain the term “plate”. Why are they called plates? Maybe an additional scheme and optical photographs of the PE plates would be helpful.

    Author’s reply: In this work, the TA/Fe3+ solution was quantificationally sprayed on the one surface of PE plates using spraying equipment. The detailed preparation process of TA/Fe3+ modified PE plates are depicted in the revised Supporting information and marked with yellow background. The PE plate is a thin and transparent PE sheet as shown in Scheme S1c. To make readers better understand it, we draw a Scheme S1, in which the photograph of PE plates and TA/Fe3+ modified PE plates as well as schematic illustration of preparation process of TA/Fe3+ modified PE plates were displayed.

2. What reagent is PI (containing Calcein AM and PI) mentioned in section 2.6.2?

    Author’s reply: AM and PI is the abbreviations of acetoxymethyl ester (AM) and propidium iodide (PI), respectively. We make some suitable corrections in the line 8-9 of section 2.1 of the revised manuscript, which explains in detail the meanings of AM and PI.

3. How do the authors explain the hysteresis on the alternation of hydrodynamic radius observed during heating and cooling step, for samples PE-I (a), PE-II (b) in Figure S4?

    Author’s reply:  We detect the hydrodynamic diameter (Dh) of both PNIPAAm2000-APTES-TA and PNIPAAm1300-APTES-TA again using Dynamic light scanning (DLS) measurement. The results are similar to the Fig. S4 a and b in the original manuscript. We think the fact that Dh of both PNIPAAm2000-APTES-TA and PNIPAAm1300-APTES-TA increase with heating, may result from the TA release. TA in PNIPAAm2000-APTES-TA and PNIPAAm1300-APTES-TA was verified to be more easily release compared with that of PNIPAAm860-APTES-TA and PNIPAAm500-APTES-TA in Section 3.3. In the heating process, for the PNIPAAm2000-APTES-TA and PNIPAAm1300-APTES-TA, TA was rapidly released into solution, the hydrogen-bonding effect that maintained the inner weak cross-linking network of nanogels was impaired, resulting in the fact that the swelling of nanogels overcame the collapse that originated from the increasing hydrophobicity of polymer chains. Thus, the curves of Dh of both PNIPAAm2000-APTES-TA and PNIPAAm1300-APTES-TA with heating look abnormal. In contrast, the TA in PNIPAAm860-APTES-TA and PNIPAAm500-APTES-TA is relative difficult to be released. So the impact of TA's release on cross-linking network is negligible. In the heating process, only the fact that the increasing hydrophobicity of polymer chains caused the collapse of nanoparticle, was displayed. So the curves in which the Dh decreases with elevating temperature were presented for PNIPAAm860-APTES-TA and PNIPAAm500-APTES-TA.

Figure S4 (the original Supporting information) Alternation of hydrodynamic radius with temperature: PNIPAAm2000-APTES-TA (a), PNIPAAm1300-APTES-TA (b), PNIPAAm860-APTES-TA (c) and PNIPAAm500-APTES-TA (d). (the first heat cycle).

    Additionally, we also make the second heat cycle for the samples I, II, III and IV. For the sample III (PNIPAAm860-APTES-TA) and IV (PNIPAAm500-APTES-TA), the curves of Dh with temperature are agreement with the first ones. But the curves of Dh with temperature for sample I (PNIPAAm2000-APTES-TA) and II (PNIPAAm1300-APTES-TA) turn into the ones of typical thermoresponsive behavior (Fig. S4 in the revised Supporting information). The results demonstrate that a new equilibrium between the released TA and nanogels is developed through suppression of the second TA release by the released TA in PBS during the first heat cycle. Thereby, in order to avoid the author’s confusion, we use the data of the second heat cycle to replace the first. The new Fig.S4 was drawn in the revised supporting information.

Figure S4 (the revised Supporting information) Alternation of hydrodynamic radius with temperature: PNIPAAm2000-APTES-TA (a), PNIPAAm1300-APTES-TA (b), PNIPAAm860-APTES-TA (c) and PNIPAAm500-APTES-TA (d). (the second heat cycle)

    In a word, according to the results of our further study, the fact that the swelling of PNIPAAm2000-APTES-TA and PNIPAAm1300-APTES-TA with more easily TA release overcame the collapse of nanogels that resulted from the increasing hydrophobicity with the increase temperature was able to explain the abnormal curves in Fig. S4a and b of the original Supporting information. We also describe the condition of more detailed swelling measurement in Section 2.4 of the revised manuscript.

4. Τhe English language should be polished. For example, throughout the main text as well as the supplementary material, words and abbreviations are mistaken (e.g. AT/Fe3+ modified PEs, Iinitail, PNIPNNm, PNIPAAm-APEST, SEM imagines).

    Author’s reply: We are really sorry about this, and the revised manuscript has been edited and corrected thoroughly. In addition, we also checked the manuscript sentence by sentence and made necessary corrections. All the changes are highlighted using yellow background. Hopefully the revised manuscript can meet the standard of publication in terms of language and grammar now.

Round 2

Reviewer 2 Report

Comments and Suggestions for Authors

The authors worked very well on the comments, the paper can be accepted in its present form.

Comments on the Quality of English Language

Minor editing of English language required.

Reviewer 4 Report

Comments and Suggestions for Authors

I see that the authors have addressed all my questions/suggestions and now their manuscripts quality and content are ammeliorated. I think that it can be published in its present form, I have no further questions to make.